Glacial allopatry vs. postglacial parapatry and peripatry: the case of hedgehogs

Černa Bolfíková Barbora bolfikova@ftz.czu.cz 1
Eliášová Kristýna 2 3
Loudová Miroslava 2
Kryštufek Boris 4
Lymberakis Petros 5
Sándor Attila D. 6
Hulva Pavel 2 7
1 Faculty of Tropical AgriSciences, Czech University of Life Sciences Prague , Prague , Czech Republic
2 Faculty of Science, Charles University in Prague , Prague , Czech Republic
3 Department of Zoology, National Museum , Prague , Czech Republic
4 Slovenian Museum of Natural History , Ljubljana , Slovenia
5 Natural History Museum of Crete, University of Crete , Heraklion , Greece
6 Faculty of Veterinary Medicine, University of Agricultural Sciences and Veterinary Medicine Cluj-Napoca , Cluj-Napoca , Romania
7 Faculty of Science, University of Ostrava , Ostrava , Czech Republic
Feng Rui
Electronic publication date: 2017 Apr 25
Publication date: 2017
Volume: 5
Electronic Location ID: e3163
Received 2016 Dec 9; Accepted 2017 Mar 8
Copyright: ©2017 Černa Bolfíková et al.
Copyright year: 2017
Copyright holder: Černa Bolfíková et al.
License: This is an open access article distributed under the terms of the Creative Commons Attribution License, which permits unrestricted use, distribution, reproduction and adaptation in any medium and for any purpose provided that it is properly attributed. For attribution, the original author(s), title, publication source (PeerJ) and either DOI or URL of the article must be cited.
License URL: https://creativecommons.org/licenses/by/4.0/

Keywords: Balkan, Founder effect, Genetic differentiation, Interspecies interactions, Landscape genetic, Phylogeography, Erinaceus roumanicus

Funding: Grant Agency of Charles University in Prague 702214 Czech University of Life Sciences Prague 20165015 The project was funded by the Grant Agency of Charles University in Prague, project no. 702214 to ML, KE, BČB, HP, and by Czech University of Life Sciences Prague, project no. 20165015 to BČB. The funders had no role in study design, data collection and analysis, decision to publish, or preparation of the manuscript.

==============================
Although hedgehogs are well-known examples of postglacial recolonisation, the specific processes that shape their population structures have not been examined by detailed sampling and fast-evolving genetic markers in combination with model based clustering methods. This study aims to analyse the impacts of isolation within glacial refugia and of postglacial expansion on the population structure of the Northern White-breasted hedgehog (Erinaceus roumanicus). It also discusses the role of the processes at edges of species distribution in its evolutionary history. The maternally inherited mitochondrial control region and the bi-parentally inherited nuclear microsatellites were used to examine samples within the Central Europe, Balkan Peninsula and adjacent islands. Bayesian coalescent inference and neutrality tests proposed a recent increase in the population size. The most pronounced pattern of population structure involved differentiation of the insular populations in the Mediterranean Sea and the population within the contact zone with E. europaeus in Central Europe. An interspecies hybrid was detected for the first time in Central Europe. A low genetic diversity was observed in Crete, while the highest genetic distances among individuals were found in Romania. The recent population in the post-refugial area related to the Balkan Peninsula shows a complex pattern with pronounced subpopulations located mainly in the Pannonian Basin and at the Adriatic and Pontic coasts. Detailed analyses indicate that parapatry and peripatry may not be the only factors that limit range expansion, but also strong microevolutionary forces that may change the genetic structure of the species. Here we present evidence showing that population differentiation may occur not only during the glacial restriction of the range into the refugia, but also during the interglacial range expansion. Population differentiation at the Balkan Peninsula and adjacent regions could be ascribed to diversification in steppe/forest biomes and complicated geomorphology, including pronounced geographic barriers as Carpathians.

Background

The consequences of geographic isolation and gene flow limitation are highly debated topics in biogeography (Coyne & Orr, 2004). Strong empirical evidence for the role of isolation in different modes of speciation has been obtained by analysis of temperate species that underwent episodes of refugial isolations in the Mediterranean region during the Pleistocene climate oscillations (Hewitt, 2000). The Balkan Peninsula acted as one of the main glacial refugium for European biota (Hewitt, 2000) and is considered as an important source of endemism and genetic diversity due to its position, historical development, topography and climatic variability (Kryštufek & Reed, 2004). After the Last Glacial Maximum (LGM), many species expanded from this area northward (Hewitt, 2000). Range expansion may be limited not just by various abiotical factors, but also by formation of a contact zone and parapatry with a related species. Interspecies interactions may lead to quick reinforcement, i.e., evolution of adaptations preventing formation of hybrids. Terrestrial taxa whose ranges are restricted by the sea barrier may colonise particular islands, providing an opportunity for peripatric evolution which is characterised by a strong founder effect, genetic drift and other island phenomena.

Hedgehogs from the Erinaceus genus are a classical example of postglacial recolonisation of the Western Palearctic (Hewitt, 2000). Recently, we recognize three species occupying this area. Northern White-breasted hedgehog (Erinaceus roumanicus (Barrett-Hamilton, 1900)) currently inhabits Central and Eastern Europe, the Baltic, and the Balkan Peninsula, and its eastern range stretches to western Siberia. The current data indicate that the Balkan Peninsula used to be the only refugium for E. roumanicus (Seddon et al., 2001; Hewitt, 2001). The Caucasus Mountains and the Bosporus strait currently form a barrier between E. roumanicus and its sister species Southern White-breasted hedgehog (E. concolor Martin, 1837), which inhabits Asia Minor. It has been estimated that these two species diverged approximately 0.4–1.4 million years ago (Bannikova et al., 2014). Western Europe and Scandinavia are populated by Western European hedgehog (E. europaeus Linne 1758) whose refugial area used to exist in the Iberian and Apennine Peninsulas. After the LGM, hedgehog species expanded from the southern refugial areas and created contact zones in the Baltics and in Central Europe (Seddon et al., 2001; Santucci, Emerson & Hewitt, 1998). Central European contact zone was probably established during the Neolithic deforestation and occurrence of interspecies hybridization is probably rare or absent (Bolfíková & Hulva, 2012). On the other hand one hybrid individual was reported from Baltic contact zone and the authors speculate that hybridization must be common in this area (Bogdanov et al., 2009). Seddon et al. (2001) suggest the presence of at least three mitochondrial lineages of E. roumanicus. The first lineage, which previously expanded from the Balkans to Russia and Ukraine, colonises the northern foothills of the Caucasus Mountains. The second lineage colonises western parts of Europe, and the third lineage is present in Greece.

A large morphologic variability has been observed for E. roumanicus throughout its wide population range, and multiple subspecies have been described. A study has recently proved that body size variability is strongly associated with the environment, and it clinally varies in the mainland (Kryštufek et al., 2009). A morphometric study by Kryštufek et al. (2009) revealed a positive correlation between body size and temperature and a negative correlation between body size and summer precipitation.

Association between particular species and their habitats give rise to the need of comparative phylogeographic studies. Dependency of Erinaceus hedgehogs on deciduous woodland (Rautio et al., 2014; Jensen, 2004) is in concordance with similar phylogeography observed in Oak and Silver Fir (Hewitt, 2000; Seddon et al., 2001) and implies that the extension of forests was followed by the hedgehogs. Some literature highlights an E. roumanicus preference for lowlands and open landscapes, particularly during the spatial expansion (Bolfíková & Hulva, 2012; Seddon et al., 2002). However, modern studies based on past environmental niche modelling are missing. Moreover, ecological considerations are based rather on knowledge of Erinaceus europaeus, while data on E. roumanicus are scarce. Different position of these two species ranges on often overlooked oceanic–continental axis, represented by more humid and less seasonal climate on the west and the opposite in the east, should be also considered (Stewart et al., 2010). The southern parts of E. roumanicus range are located mainly in drier biomes in the Mediterranean Basin and the Pontic-Caspian steppe. The northern parts of the range are situated mainly in the temperate deciduous forests. Adaptations to the more dry open and seasonal habitats in E. roumanicus could not be excluded (Bolfíková & Hulva, 2012).

Do the classical models have a potential to further develop phylogeogeographic paradigm using up to date tools of the landscape genetics? Here, we provide new information on the hedgehog’s evolutionary history using a combination of fast evolving markers, including nuclear microsatellites and mitochondrial control region. By analysing multiple E. roumanicus samples ranging from the post-refugial area in the Balkan Peninsula to the parapatric contact zone with E. europaeus in central Europe, we aim to:

(1) assess the population structure of E. roumanicus within the studied area using the spatial and non-spatial Bayesian clustering methods and discuss observed patterns within landscape genetic framework in the light of geomorphology and habitat quality of the region;

(2) reveal the history of island populations and suggest possible way of colonisation of these islands. He hypothesize that insular effects like the founder effect, genetic drift and other factors which decrease genetic variation highly affected these populations;

(3) analyse the populations in the secondary contact zone from Central Europe. We hypothesize that respective populations were affected by species interactions, including introgression and reinforcement;

(4) analyse the populations from the post-refugial areas at the Balkan peninsula and assess the hypothesis of occurrence of the glacial subrefugia.

Material and Methods

Sampling and genotyping

Samples obtained from muscle tissue or digits or ears were collected from road-killed animals or from museum collections. No special permit is needed from the Institutional Review Board. The tissues were fixed in 96% ethanol and stored at −20 °C. The samples were obtained in cooperation with the Natural History Museum of Crete (Petros Lymberakis), the Slovenian Museum of Natural History (Boris Kryštufek), and the University of Agricultural Sciences and Veterinary Medicine Cluj Napoca (Attila Sandor). For the study, 314 samples were analysed from the following European countries: Czech Republic (49), Slovakia (26), Hungary (9), Romania (43), Slovenia (44), Italy (2), Croatia (20), Bosnia and Herzegovina (25), Montenegro (6), Serbia (10), Bulgaria (4), Macedonia (17) and Greece (59), which included several islands (the Ionian islands, Euboea, Cyclades, and Crete). Each sample’s origin, GPS coordinates, and date of collection along with the name of the person who collected it are listed in Table S1. A map (Fig. 1), indicating the geographic origins of the samples, was created using Geographic Information System (ESRI, 2014).

Figure 1 Sampling localities for the 314 samples of the Northern White-breasted Hedgehog (Erinaceus roumanicus) in central Europe and the Balkan region.

Each sampled individual is represented by a pink dot. The distribution map of the species is shown in the smaller square. Map was done using ArcGIS (ESRI, 2014) using publicly available layers (Bathymetry-EMODnet, Administrative areas-GADM database, Elevation-WorldClim database, Rivers-EEA hydrographic data set). Inset of the figure is based on an IUCN distribution map of species (Amori et al., 2016) edited for the purposes of this article in Photoshop CS3 (Adobe).

DNA was extracted using a commercial DNA Blood and Tissue Kit (Qiagen, Hilden, Germany). A combination of nine microsatellite markers and mtDNA control region sequences was used. Laboratory procedures were performed according to the protocols by Bolfíková & Hulva (2012), details may be found in Table S3. Each dataset was analysed separately as specified below.

Non-spatial analyses of genetic variability

The sizes of the microsatellite alleles were determined using GeneMarker V1.85 (http://www.softgenetics.com). Data binning was performed with the AutoBin software (http://www4.bordeauxaquitaine.inra.fr/biogeco/layout/set/print/Ressources/Logiciels/Autobin). The presence of the null alleles (false homozygotes), stutter bands and large allele dropouts was tested using Micro-Checker (Van Oosterhout et al., 2004). Bayesian statistics for recognising hybrid categories was run in NewHybrids 1.1 (Anderson & Thompson, 2002).

Evaluation of the genetic structuring of the microsatellite dataset was performed with a Bayesian Clustering Analysis in Structure 2.2 (Pritchard, Stephens & Donnely, 2000). Structure determines the most likely assignment of individuals to clusters (K) based on analysis of likelihood. Structure is powerful to detect clinal variability and high geographic admixture in the dataset (Chen et al., 2007). The parameters of the final run were as follows: 1,000,000 MCMC steps, with burn-in-period 100,000 and 5 iterations for each K. An Admixture Model of Correlated Frequencies was used. The number of tested population was within the K = 1–14 interval. The results of the analyses were combined and evaluated by Structure Harvester (Earl & VonHoldt, 2012) using the Evanno method (Evanno, Regnaut & Goudet, 2005). Graphic visualisation was provided using Distruct 1.1 (Rosenberg, 2004). The inbreeding coefficient (FIS) for each population was tested by GenePop (Rousset, 2008). Departures from the Hardy-Weinberg equilibrium (HWE), number of alleles (Na) and allelic richness (AR) were estimated in FSTAT (Goudet, 1995). Allelic richness was used to estimate an allele numbers, across all loci, corrected for equal sample size in all analysed populations. The population with the smallest number of individuals was N = 24 from Crete. The expected (HE) and observed (HO) heterozygosities were computed using Genetix (Belkhir et al., 1999).

Sequences of the mtDNA partial control region were edited using Geneious (Kearse et al., 2012) and aligned using MAFFT (Katoh & Standley, 2013). Particular haplotypes were identified in DnaSP v.5 (Rozas et al., 2003). The haplotype data were submitted to GenBank® (Accession numbers are from KY489901 to KY489953). The relationships among the haplotypes were visualised by the median-joining network (Bandelt, Forster & Röhl, 1999) using Network 4.5 (http://www.fluxus-engineering.com). The demographic parameters were estimated by summary statistics of the genetic variability for each population (recognised by the Bayesian clustering analysis of the haplotype data using the Geneland software; details in spatial analyses below). Tests for haplotype diversity (h; represents the probability that two randomly chosen haplotypes are different), nucleotide diversity (π; the average number of nucleotide substitutions per site between two sequences) and neutrality tests (Tajima’s D, Fu’s FS and R2) were computed for each population in DnaSP v.5 (Rozas et al., 2003). Fu’s FS(for big sample sizes) and R2 (for small sample sizes) are considered as the most powerful tests (Ramos-Onsins & Rozas, 2002; Ramírez-Soriano et al., 2008). Significantly negative results indicate presence of selective sweep or strong population growth signal in the data. The course of the past population size was estimated with coalescent-based Bayesian skyline plots (Drummond et al., 2005). We inferred the model of sequence evolution in jModelTest (Posada & Crandall, 1998) using Bayesian information criterion (BIC). The Bayesian skyline plot analysis was conducted using the BEAST 1.4.8 program (Drummond & Rambaut, 2007) with a strict molecular clock and a GTR+I+G substitution model. The MCMC procedure was run three times for each species with 30,000,000 iterations, and the genealogy and parameters of the model were stored every 1,000 iterations. The results were combined in LogCombiner, and burn-in was set to 10,000,000 iterations in each run. The Bayesian skyline plot and the convergence of chains were assessed by Tracer v.1.4 (Rambaut et al., 2014).

Spatial analyses of genetic variability

To analyse the spatial genetic architecture of E. roumanicus in the area of interest, we used two spatial model based clustering approaches, including software Geneland 4.0.4 (Guillot, Mortier & Estoup, 2005) and TESS (Chen et al., 2007; Durand et al., 2009). Geneland encompasses Bayesian clustering of individual genotypes based on separation of random mating subpopulations and produces Dirichlet cells centred on subpopulation territories. Geneland can analyse both haplotype data and codominant data in independent runs. The parameters for the Geneland analysis were set for each marker type as follows: 500,000 MCMC iterations, thinning = 100, five independent runs and K = 1–15. The posterior probability of the subpopulation membership for the most supported run was computed for each pixel of the spatial domain, using a burn-in of 50 iterations. An uncorrelated Frequency Model was chosen. TESS is based on minimizing of Wahlund effect and produces Dirichlet cells based on individuals and is using microsatellite data. We chose BYM model due to lower Deviance Information Criterion (DIC) and higher stability of algorithm. The first run of program was set to Kmax = 2–10 populations, then we plotted Kmax versus DIC values and selected the optimal Kmax (see details in Fig. S1). The final run of TESS was set to Kmax = 5–6, with 50,000 sweeps, a burn-in 30,000, the interaction parameter ϕ = 0.6 and 100 iteration for each K. The lowest DIC values were performed for Kmax = 6 and we chose as representative the run within 30% of runs with lowest DIC values, displaying five colours.

Combination of these approaches may be useful in exploring spatial population structure (Balkenhol et al., 2015), as Geneland performs well in detecting recent linear barriers to gene flow (Blair et al., 2012), while TESS addresses situations with barriers of complex shapes and permeability for migrants, as well as secondary fusions of subpopulations (Chen et al., 2007).

The Mantel test (Mantel, 1967) was used to characterise the presence of isolation-by-distance to evaluate the correlation between congruent similarity/dissimilarity matrices, which addressed the relationship between the geographic and genetic distances of observations across a landscape provided by the Alleles in Space (AIS) software (Miller, 2005). The number of iterations used was 1,000. We used the AIS software for the Genetic Landscape Shape Analysis to compute the genetic distances, which were assigned to the connector centres of each locality using Delauney’s triangulation. The interpolation procedure, which determined the presumed genetic distances in a lattice frame containing 15-km units, covered the area of interest, and a graph of the interpolated distances was created. Peaks in this graph represent an area of high genetic distances and should be correlated with areas of restricted gene flow. Both marker types were independently analysed using AIS software.

Results

We did not detect any genotyping errors that resulted from a large allele dropout or stutter bands in 260 samples that were successfully genotyped at nuclear markers. All genotyped individuals are indicated in Table S1. Micro-Checker detected null alleles in several microsatellite loci, and the estimated frequencies differed between loci and populations. The estimated frequencies were lower than 0.2432 for all populations examined. In the Cretan population, two loci, EEU12H (0.2432) and EEU3 (0.2099), showed high null allele frequencies using the methods described by Van Oosterhout et al. (2004). Primers were designed for E. europaeus (Becher & Griffiths, 1997; Henderson et al., 2000) and tested by Bolfíková & Hulva (2012). All genotypes were originally obtained from the tissues, which are less susceptible to the presence of null alleles than non-invasive samples. Further analyses proved that the Cretan population was influenced by the founder effect, genetic drift and inbreeding. Thus, the null allele estimation was probably influenced by the high homozygote frequency in the population.

Sequences produced by Sanger sequencing were of good quality and some individuals with ambiguous sites were re-sequenced or skipped from the analyses. A total of 296 partial control region sequences (404 bp) were obtained and represented 53 haplotypes among the analysed samples (see details in Table S1).

Non-spatial analyses of genetic variability

The nine selected microsatellite loci were polymorphic. The total number of alleles in a single locus varied from 5 to 23 for the entire dataset. One sample from the Slovakia (Trencianske Teplice) was omitted from further analysis because it was suspected of having a hybrid origin. A follow-up analysis with the NewHybrids program (Anderson & Thompson, 2002) using the original dataset from Bolfíková & Hulva (2012) confirmed that the sample was a backcrossed hybrid of E. europaeus and E. roumanicus; the mitochondrial DNA originated from E. roumanicus.

From the total number of tested clusters (K = 1–14), division into the three populations (K = 3) obtained the highest support according to Evanno, Regnaut & Goudet (2005). The populations were differentiated as follows: (i) samples from Crete and from the Hellenic peninsula in southern Greece; (ii) samples from the Balkan area (Serbia, Bulgaria, Macedonia, Greece, Montenegro, Croatia and Bosnia and Herzegovina) plus Slovenia and Romania; and (iii) samples from Central Europe (Czech Republic, Slovak Republic and Hungary) (Fig. 2A). The second highest ΔK value was observed for a division into the seven populations (K = 7). An increasing number of clusters resulted in further differentiation of the Balkan population by separation of the population from Pannonia (at K = 4), differentiation of the Slovenian population (K = 5) and separation of populations from the west and east of the Balkans (from K = 6). However, there is certain degree of admixture among particular clusters. Two different landscape genetic algorithms supported similar spatial pattern with three (Geneland, Fig. 5D) and five (TESS, Fig. 2B) clusters.

Figure 2 Structuring of the nuclear data.

(A) Population assignment using Bayesian clustering analysis in Structure for the 260 Erinaceus roumanicus samples in Central Europe and the Balkan region. The results are shown for K = 2 to K = 7. The highest ΔK was for K = 3, and the second highest ΔK was for K = 7. (B) Geographical cluster assignments implemented by TESS. The results display five colours for Kmax = 6.

The aforementioned differentiation was also tested without the Czech and Slovakian samples, whose genetic architecture could have been influenced by phenomena associated with the recently developed contact zone. This analysis of Structure supported the previous subdivision of our dataset.

The genetic variability parameters are given for the seven populations (Table 1) and three populations (Table S2). HO was significantly lower than HE and GenePop analyses showed a significant lack of heterozygotes (p < 0.01) for each population. Crete had the lowest AR and HE. The highest FIS was detected in Romanian population.

Table 1 The descriptive statistics of microsatellite genetic diversity in seven populations of Erinaceus roumanicus, which were recognised by the Bayesian clustering analysis using Structure.

Number of alleles (Na), allelic richness (AR), expected heterozygosity (HE), observed heterozygosity (HO), inbreeding coefficient (FIS) are given for each population.

	Na	AR	HE	HO	FIS	
Czech Republic	7	6.187	0.7120	0.6151	0.134	
Slovakia, Hungary	9	7.318	0.7032	0.6291	0.097	
Romania	9	6.575	0.7521	0.6171	0.242	
Slovenia	8	6.317	0.6844	0.6071	0.102	
Croatia, Bosnia and Herzegovina	9	7.045	0.7125	0.6442	0.114	
Bulgaria, Greece, Macedonia, Serbia, Montenegro	10	7.912	0.7628	0.6923	0.102	
Crete	6	4.994	0.5243	0.4292	0.167	

The median-joining network didn’t demonstrate reciprocally monophyletic lineages in the studied area (Fig. 3). Haplotypes are shared across populations, but some trends are visible. One unique haplotype shared with the majority of individuals from Crete and a few other haplotypes carried by one or two individuals are representing the population of South Balkan (in blue on Fig. 3). Similar pattern is representing Central Europe (in orange on Fig. 3). The North Balkan population (in pink on Fig. 3) has two locally common haplotypes, but individuals from Central Europe and the Dinaric area are also share them. The Carpathian population (in yellow on Fig. 3) had several haplotypes in common with other populations but also several unique haplotypes. Samples from Dinaric area (in green on Fig. 3) were wide-spread in the network and produced star-like pattern around haplotype that is common in Slovenia, Hungary and Croatia. Black sea coast population (in violet on Fig. 3) has several unique haplotypes which are widespread within the network.

Figure 3 Median-joining network of the mitochondrial control region haplotypes for 296 Erinaceus roumanicus samples.

Haplotypes are represented by circles, which are proportional to the haplotype frequency. The hypothesized haplotypes are represented by red dots. Colour codes are used according to the population assignments of the samples using mtDNA data from the Geneland analysis (see Fig. 5C or Table S1). Lines between nodes represent a nucleotide changes.

The genetic variability parameters for mitochondrial data were estimated for six populations according to the results of analysis in Geneland (Table 2). All tests of neutrality yielded negative values indicating recent population growth, only Central European (containing Czech Republic, East Slovakia and North Hungary) population neutrality tests were not significant (Table 2). The Bayesian skyline plot indicated a long period with a stable population size and its recent increase (Fig. 4).

Table 2 The summary statistics of the partial control region of mitochondrial DNA in six populations of Erinaceus roumanicus, which were recognised by Bayesian analysis using Geneland.

Colours in bracelets correspond to colours used in Fig. 5C. Number of individuals (N), number of haplotypes (Nh), haplotype diversity (Hd), number of polymorphic sites (Np), nucleotide polymorphism (π), Tajima’s D (D), Fu’s Fs (FS) and R2 are given for each population and for whole dataset.

	n	Nh	Hd	Np	π	D	R2	FS	
Central Europe (orange)	58	6	0.3134	10	0.00204	−0.92848	0.0688	−0.561	
Carpathians (yellow)	41	10	0.8269	9	0.00292	−0.81891	0.0874	−2.779*	
North Balcan (pink)	63	15	0.6912	17	0.00272	−1.35223	0.0534	−7.052***	
Black sea cost (violet)	23	9	0.7787	14	0.00548	−1.25805	0.0817*	−2.731	
Dinarids (green)	59	17	0.8954	19	0.00568	−0.90052	0.0730	−5.470**	
South Balkan (blue)	52	11	0.3808	18	0.00229	−1.82115*	0.0492*	−5.937*	
All data	296	53	0.9185	38	0.00644	−1.38337	0.0415	−3.638***	
Notes. The significant values tests are shown with asterisks:

* p < 0.05.

** p < 0.01.

*** p < 0.001.

Figure 4 Bayesian skyline plot based on mtDNA of 296 Erinaceus roumanicus samples.

The X axis represents the time in mutation units per nucleotide position. The Y axis represents a correlate of the population size (Ne). The black line shows the median of the Ne estimation, and the 95% confidence intervals are indicated by the blue areas.

Spatial analyses of genetic variability

The Mantel test indicated a significant isolation by distance (r = 0.303; p < 0.001) within our dataset. Geneland analysis of microsatellite markers determined that the division of our dataset into three populations (Central Europe, mainland area, and south Greece plus islands) was the most likely to occur (Fig. 5D). Analysis implemented by TESS revealed the most fitting Kmax = 6, which spatially represents five clusters corresponding to population from Crete, Czech Republic, Pannonian area, Eastern and Western part of Balkan Peninsula (Fig. 2B). This pattern is in concordance with results of Structure analysis for K = 5 (Fig. 2A). Mitochondrial analysis divided our dataset into the following six clusters: (i) South Balkan (southern and central Greece, Peloponnese, Crete and the adjacent islands); (ii) Dinaric area (north Greece, Macedonia, south Bosnia and Hercegovina, south Croatia, Montenegro and south Serbia); (iii) Black sea coast (Thrace and most of Bulgaria and southeast Romania); (iv) Carpathian area (northwest Romania, east Slovak Republic and east Hungary); (v) North Balkan (north Serbia, north Bosnia and Herzegovina, north Croatia, Slovenia, southwest Hungary and north Italy); and (vi) Central Europe (northwest Hungary, west Slovak Republic and Czech Republic) (Fig. 5C; Table S1). Genetic Landscape Shape Analysis using AIS software detected very low genetic distances in Crete and the Czech Republic based on the mitochondrial data and nuclear data in particular. However, high genetic distances were observed with the microsatellites among samples in regions to the south and east of the Balkan Peninsula, with the highest peaks observed in Romania (Fig. 5B). The mitochondrial data showed flat genetic landscapes (Fig. 5A).

Figure 5 Genetic landscape shape interpolation analysis.

Genetic landscape shape interpolation analysis based on variation of (A) the control region of mtDNA using 296 Erinaceus roumanicus samples and (B) nine nuclear microsatellite loci using 260 E. roumanicus samples. The X and Y axes reflect the geographic coordinates within the study area, while the Z axis represents the average genetic distances between the analysed samples. The Bayesian landscape genetic analysis using Geneland illustrates the spatial distribution of the populations using (C) the control region of mtDNA and (D) nine nuclear microsatellite loci. Figure was done using publicly available outlines (https://CRAN.R-project.org/ package=rworldmap) and edited in Photoshop CS3 (Adobe).

Discussion

Based on the results obtained using both non-spatial and spatial individual based clustering approaches, the population from the southernmost region of the range (including southern Greece and the adjacent islands) and the population from the contact zone with E. europaeus were separated from the rest of the range. This pattern indicates that processes acting at peripatric and parapatric range edges shaped the genetic architecture of the species substantially. Further differentiation in clustering hierarchy provides evidence for occurrence of subpopulations within southeast part of the range. This result could be ascribed to complicated geomorphology and land cover of the region and may imply also subdivision of refugial populations in the past.

Insular populations in the Mediterranean Sea

All analyses indicate strong differentiation of the individuals from Crete, Cyclades and from Peloponnese, Euboea and southern and central Greece. As the largest island of Greece and the fifth largest Mediterranean island, Crete is typified by impoverished native mammalian fauna and the high impact of recently introduced species. However, Crete is also known as a generator of novelty in the numerous lineages that were able to colonise the island. From a macroevolutionary viewpoint, Crete is the place where pygmy forms of deer, elephants and hippopotamuses occurred (Van der Geer et al., 2010). From the microevolutionary viewpoint, the cryptic bat species occur there (Hulva et al., 2007; Hulva et al., 2010). Peloponnesus is also typified by its cryptic variability in vertebrates (Gvoždík et al., 2010). The isolation of this region was recently enhanced by the 1893 construction of the Corinth Canal, which effectively separated Peloponnesus from the mainland. However, our analyses do not confirm a reciprocal monophyly for the E. r. nesiotes subspecies, which supports the findings by Schaschl, Lymberakis & Suchentruck (2002). Moreover, we ascertain that in addition to Crete, the range of this population includes other islands and reaches continental Greece. The assessment of the geographic origins and dispersal scenarios of this population will require more detailed analysis. However, the presumably absent land bridge between Crete and the mainland during the Pleistocene period (Schule, 1993), the absence of hedgehogs in Crete’s fossil record (Lax & Strasser, 1992) and the frequent introductions of hedgehogs onto the islands by humans (Bolfíková et al., 2013) indicate that the role of the anthropogenic factor in the phylogeography of the southern cluster is highly probable.

The first archaeological evidence of hedgehogs in Crete was collected from an Early Minoan tomb (Jarman, 1996), but the author concluded that the hedgehog might have been an intrusion (e.g., entered the tomb much later). Using allozymes and partial cytochrome b sequences, Schaschl, Lymberakis & Suchentruck (2002) proposed that hedgehogs were introduced to Crete by archaic settlers from the mainland. This population was previously described as a subspecies, E.r. nesiotes, by Dorotha Bate in 1906 based on phenotypic data. These facts imply the occurrence of insular syndrome in the Cretan population. The markedly decreased genetic variation values in the insular population imply that factors such as the founder effect and genetic drift played substantial roles. The agreement between the genetic variation patterns and the phenotypic data indicates that insular syndrome, including dwarfism in hedgehogs from the Greek islands, has occurred (Kryštufek et al., 2009; Giagia-Athanasopoulou & Markakis, 1996). Despite the uncertainty surrounding the origins of differentiation for the southern cluster, the aforementioned results indicate a pronounced role for peripatric processes in the microevolution of E. roumanicus. Elevated evolutionary rates in islands are obvious also in phenotypic evolution. The trends in body size changes of insular hedgehogs are negatively correlated with the island’s distance from the mainland, however, the insular response was not uniform (Kryštufek et al., 2009).

Secondary contact zone with E. europeaus in Central Europe

The population from Central Europe is genetically different from the rest of the mainland, as suggested by all Bayesian clustering analyses. This cluster is also characterised by its lower genetic variation, even though, in some parameters still comparable to post-refugial populations. This may be the outcome of spatial expansion, which is usually followed by a decrease in genetic variability due to the bottleneck effect. More complex phenomena, such as allele surfing (increase of an allele frequency in the wave front of an expanding population), might alter the genetic architecture of the population’s expanding edge and change the genetic variation as well (Klopfstein, Currat & Excoffier, 2006; Excoffier & Ray, 2008; Excoffier, Foll & Petit, 2009). Demographic analyses did not confirmed statistically significant population size increase. Thus, we suspect that the aforementioned phenomena played important roles in shaping the population genetics of Central European hedgehogs. Other factors that caused population differentiation in this case may be associated with parapatry and its consequent species interactions. Ecological interactions and their consequent selections may have influenced the diversity of the studied microsatellite loci by genetic hitchhiking, while hybridisation and introgression in the secondary contact zones may have further shaped the allopatrically evolved lineages and potentially cover the signal of recent population expansion. These potential factors will require further study and genomic tools.

We are the first to report a single hybrid sample from the range margins of E. europaeus in Central Europe (Slovakia—Trenčianske Teplice) using traditional genetic markers. Interspecies interactions in the sympatry zone may have previously played roles in forming reproductive barriers. We hypothesised that hybridisation occurred when the two species came into contact for the first time, when the isolating mechanisms were not yet formed. Afterwards, during the reinforcement phase, species developed mechanisms for reproductive isolation and the hybrid event frequency decreased for follow-up generations. The history of possible hybridisation events in the Central European population should be tested using approaches that include whole genome data because the introgression modes may be complex. Future investigations should address the contact zone at the Slovenian/Italian border (located to the south of the Alps) and the effect of altitudinal heterogeneity on both species. This region involves the contact zone for both species and the transition zone between the Slovenian mountains and the Padan Plain.

The lower genetic diversity associated with the range margins and the novel selective pressures applied by interactions with related species may be why the Central European population was highly divergent from the other populations in the dataset.

Post-refugial areas at the Balkan Peninsula and adjacent regions

The separation of the Pannonian population represents the most pronounced pattern after the split of parapatric and peripatric populations. The Pannonian basin is a confined area from the view of geographical isolation by the Carpathian arch and vegetation cover typified by grass steppe, i.e., it possess habitat similarity with the part of species range in Pontic-Caspian steppe. Consequently, it represents a pronounced biogeographical region of Europe and a presumed continental interglacial refugium of continental and steppe faunal elements (Stewart et al., 2010; Ricanova et al., 2011). In E. roumanicus, this pattern might mirror geographical isolation, ecological differentiation, and also the quality of ancestral habitat and larger extent of steppe biome during glacial periods.

Further differentiation occurs between populations from the west of the peninsula at the Adriatic coast and the east at Black Sea coast. This differentiation might be ascribed to isolation by distance and barrier role of Carpathian Mountains. For example, based on the results provided by AIS, the largest genetic distances between the samples in our dataset are located in Romania. This pattern implies a restricted level of gene flow within the population, which might be ascribed to the high altitudinal heterogeneity of the Carpathian Mountains. However, this pattern may also mirror occurrence of forest refugia at Adriatic and Pontic regions. The Black Sea shore likely served as a glacial refugium for multiple animal species, such as Bombina bombina (Fijarczyk et al., 2011) and Emys orbicularis (Joger et al., 2007). The presence of trees during the LGM has also been proven (Tzedakis, 2004). Palynological analyses have confirmed occurrence of isolated forests within the refugia in response to climatic and spatial variabilities (Huntley, 1999), for example in the Pindos Mountains in northwestern Greece, where levels of precipitation were higher (Tzedakis, 2004). In the case of the European oak Quercus, two separate Balkan subrefugia (Greece and the west coast of the Black Sea) have been located (Brewer et al., 2002). King & Ferris (1998) discovered unique haplotypes of the European alder (Alnus glutinosa) in the regions around Bulgaria and Greece. Based on these examples, Tzedakis (2004) presumed the existenceof temperate forests during the last glacial maximum in the mountainous and coastal areas, whereas two previously identified subrefugia occurred in western Greece and in eastern Bulgaria near the Black Sea.

The population from Slovenia is separated from the Balkan genotypes in Structure analyses. This pattern suggests a gene flow limitation due to the geographical isolation, caused by the Alps, the Dinarids and/or specific climatic conditions. For example, a fully grown forest was present during the LGM in Slovenia (Tzedakis, 2004). Another hypothesis suggests a possible species interaction with E. europaeus and, consequently, a local parapatic evolutionary process (see above).

Conclusions

The most pronounced pattern within our dataset was represented by the differentiation of populations at the edges of the recent range. This study indicates that peripatry and parapatry might be not only limiting factors to range expansion, but may also provide strong microevolutionary forces that shape the genetic structure of the species. Traditionally, population differentiation in temperate species has been ascribed to allopatric processes during glacial periods. Here, we provide an alternative example, showing that population differentiation may occur not only during the glacial restriction of the range into refugia, but also during the phase of interglacial range expansion. During the interglacial periods, peripatric modes of evolution for many taxa might be more frequent due to the rise in sea level, which cause increased shelf island formation. Parapatric processes are accelerated by the range expansions of lineages isolated in the refugia and by the formation of potential suture zones.

Recent population of E. roumanicus in post-refugial area at Balkan Peninsula and adjacent regions shows differentiation with particular subpopulation located in areas of interglacial steppic refugia and glacial forest refugia. A detailed habitat model and past niche modelling will be necessary to infer, which type of habitat was crucial for survival of E. roumanicus during glacial periods. However, certain degree of admixture among particular clusters is obvious, mirroring complex geomorphology and shape of geographical barriers, as well as altitudinal heterogenetity within Balkan Peninsula. The situation might be further complicated by the fact, that a mosaic of particular biomes often existed in refugia (Huntley & Birks, 1983), increasing the proportion of ecotones, and the biotas were often “disharmonious”, i.e., particular ecosystems often contain species that inhabit different habitats today (Tyrberg, 1991).

Supplemental Information

Table S1 Table of all used samples

Given are: country of origin, haplotype code, colour code in Geneland analysis, sucess of microsatellite gonotyping, used code for individual, coordinates and collectors.

Click here for additional data file.

Table S2 The descriptive statistics of microsatellite genetic diversity in three populations of Erinaceus roumanicus, which were recognised by the Bayesian analysis using Structure

Number of alleles (Na), allelic richness ( AR), expected heterozygosity ( HE), observed heterozygosity ( HO), inbreeding coefficient ( FIS) are given for each population.

Click here for additional data file.

Table S3 Table of microsatellite loci and control region primers

Table of microsatellite loci (Locus), their accession numbers (GenBank accession no.), annealing temperatures (Annealing temp.), fluorescent labels and repeat motifs. For control region, primers and annealing temperature are given.

Click here for additional data file.

Figure S1 Sizes of microsatellite alleles

Genotypes of 260 individuals

Click here for additional data file.

We would like to thank all researchers and institutions which provided samples for this study (all are listed in Table S1). We thank to P. Matějů for help in laboratory.

Additional Information and Declarations

Competing Interests

Author Contributions

Animal Ethics

DNA Deposition

The authors declare there are no competing interests.

Barbora Černa Bolfíková conceived and designed the experiments, performed the experiments, analyzed the data, contributed reagents/materials/analysis tools, wrote the paper, prepared figures and/or tables, reviewed drafts of the paper.

Kristýna Eliášová performed the experiments, analyzed the data, contributed reagents/materials/analysis tools, wrote the paper, prepared figures and/or tables, reviewed drafts of the paper.

Miroslava Loudová performed the experiments, analyzed the data, contributed reagents/materials/analysis tools, prepared figures and/or tables, reviewed drafts of the paper.

Boris Kryštufek, Petros Lymberakis and Attila D. Sándor contributed reagents/materials/analysis tools, reviewed drafts of the paper.

Pavel Hulva conceived and designed the experiments, contributed reagents/materials/analysis tools, wrote the paper, reviewed drafts of the paper.

The following information was supplied relating to ethical approvals (i.e., approving body and any reference numbers):

Tissues were collected from road-killed animals or from museum collections. No special permit is needed from the Institutional Review Board.

The following information was supplied regarding the deposition of DNA sequences:

Sequence data are stored in Genbank—accession numbers KY489901–KY489953. Microsatellite data are included in File S1.

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
