# Peer review of "Glacial allopatry vs. postglacial parapatry and peripatry: the case of hedgehogs"

_PeerJ, doi:10.7717/peerj.3163_

## Round 0.1 · original submission · Minor Revisions

Both reviewers applauded your efforts and contributions. They also requested some clarifications, which I believe will enhance the manuscript quality.

Reviewer 1 ·

Basic reporting

.

Experimental design

.

Validity of the findings

.

Additional comments

The study used genetic markers (mitochondrial control region and nuclear microsatellites) to analyze the impacts of isolation within glacial refugia and of postglacial expansion on the population structure of the Northern White-breasted hedgehog. The main contribution is that they detected the pattern of population structure in the Central Europe, Balkan Peninsula and adjacent islands and showed population differentiation may occur not only during the glacial restriction of the range into the refugia, but also during the interglacial range expansion. However, there are several minor concerns,

(1) A clearer description of experimental design is needed, such as (a) a precise introduction of samples in experiments (the numbers in lines 108-111 and 141 are not consistent and confusing) and (b) more information of genotyping markers. Uploading probe sequences of microsatellite and mtDNA markers are suggested.
(2) Because the number of loci and sample size are limited in this study, it is strongly suggested to justify the power of in performed genetic distance analyses performed in this study. Fig.5d may show insufficient power of 9 microsatellite loci.
(3) Please clarify how microsatellite and mtDNA markers were combined in the analyses (line 117). Were they combined together during the analyses? Or did authors intend to make specific conclusions from each aspect of maternal and biparental inheritance? Or did authors try to evaluate the population structure from two sets of genetic markers and support each other?
(4) The authors tried to answer multiple geomorphology questions based on hedgehog’s genetic loci. More analyses from different angles (such as fossil morphology, etc.) if applicable will provide more strong evidence.

In summary, this is an interesting study on Glacial allopatry vs. postglacial parapatry and peripatry from hedgehogs which is well known by its postglacial recolonization. The main concern is the power sufficiency.

Annotated reviews are not available for download in order to protect the identity of reviewers who chose to remain anonymous.

·

Basic reporting

no comments

Experimental design

Rational of genetic analysis needs be specified.
1. Only 9 STR markers were selected and analyzed in this study, is there a rational for marker selection?
2. Which tissues were DNA extracted from? How good DNA quality of each sample was?

Validity of the findings

Lacks of independent validation:
Line 210-211, “All genotypes were…the present of null alleles”, this statement seems not correct. How reliable the genotyping planform was? Null alleles could simply be genotyping errors, sanger sequencing is an option to validate problematic calls.

Additional comments

Bolfikova and colleagues presented an interesting evolutionary study of the northern white-breasted hedgehogs across multiple European countries. The major contribution is to illustrate interglacial range expansion could be different source for population differentiation (especially in hedgehogs). The authors also identified an inter-species hybrid of hedgehogs in center European. This study is a well-designed report, genotyping and data analysis were conducted properly. However, lacks of some technical details prevents readability of this manuscript.
My other comments as below:
1. Line 214-216, how haplotype data was generated?
2. Line 151, should explain the meanings of haplotype diversity, nucleotide diversity, and neutrality test.
3. Check with Journal’s policy about data deposition.

---

## Round 0.2 · Minor Revisions

Please address the last comment from reviewer 2.

Reviewer 1 ·

Basic reporting

All questions have been addressed and it is acceptable.

Experimental design

no comment

Validity of the findings

no comment

·

Basic reporting

No comments.

Experimental design

No comments.

Validity of the findings

No comments.

Additional comments

My previous comments have been addressed adequately. Details of sequencing and re-sequencing should be mentioned in the manuscript. For sequence data generated from this study, GeneBank Accession numbers should be provided for public access.

---

## Round 0.3 · accepted · Accept

All previous concerns have been adequately addressed.

·

Basic reporting

no comment

Experimental design

no comment

Validity of the findings

comment

Additional comments

I have no further comments based on the revised manuscript.